# Semi-Site-Specific Primer PCR: A Simple but Reliable Genome-Walking Tool

Cheng Wei [1,2], Zhiyu Lin [1,2,3], Jinfeng Pei [1,2], Hao Pan [1,2,3] and Haixing Li [1,2,*]

[1] State Key Laboratory of Food Science and Technology, Nanchang University, Nanchang 330047, China
[2] Sino-German Joint Research Institute, Nanchang University, Nanchang 330047, China
[3] School of Chemistry and Chemical Engineering, Nanchang University, Nanchang 330031, China
* Correspondence: hxli@ncu.edu.cn

**Abstract:** Genome-walking has been frequently applied to molecular biology and related areas. Herein, a simple but reliable genome-walking technique, termed semi-site-specific primer PCR (3SP-PCR), is presented. The key to 3SP-PCR is the use of a semi-site-specific primer in secondary PCR that partially overlaps its corresponding primary site-specific primer. A 3SP-PCR set comprises two rounds of nested amplification reactions. In each round of reaction, any primer is allowed to partially anneal to the DNA template once only in the single relaxed-stringency cycle, creating a pool of single-stranded DNAs. The target single-stranded DNA can be converted into a double-stranded molecule directed by the site-specific primer, and thus can be exponentially amplified by the subsequent high-stringency cycles. The non-target one cannot be converted into a double-strand due to the lack of a perfect binding site to any primer, and thus fails to be amplified. We validated the 3SP-PCR method by using it to probe the unknown DNA regions of rice hygromycin genes and *Levilactobacillus brevis* CD0817 glutamic acid decarboxylase genes.

**Keywords:** genome-walking; semi-site-specific primer; partial annealing; rice; *Levilactobacillus brevis* CD0817





## 1. Introduction

Genome-walking is a series of technologies that are exploited to retrieve unknown DNA sequences flanking a known region. Genome-walking is one of the worthy tools in many fields of life science [1,2]. By identifying unknown flanking regions, genome-walking has successfully realized a variety of applications, such as obtaining the full length of a gene, accessing the regulatory sequence of a gene, and identifying transgenic DNA [3–6].

Numerous strategies have been proposed for genome-walking. Screening the genomic DNA library is the earliest genome-walking method. This method, however, is rather frustrating because it is laborious and time-consuming [7]. With the proceeding of biotechnology, many PCR-based genome-walking methods have been devised. These PCR-based methods vary substantially in their experimental operations, but can be divided into three categories according to the involved rationales: (i) inverse PCR [8,9]; (ii) digestion-ligation-mediated PCR [10,11]; and (iii) randomly primed PCR [12,13].

Inverse PCR features high specificity and low efficiency. In inverse PCR, genomic DNA is firstly digested by an endonuclease that has no recognition site on the known sequence; the digested DNA is then self-cyclized by a DNA ligase. The cyclized target product is subsequently amplified using two site-specific primers with an orientation of 5′-end-facing-5′-end (opposite to classic 3′-end-facing-3′-end). Due to the use of a site-specific primer pair, the amplification specificity of inverse PCR is high [8,9,14]. However, the length of the fragment captured by inverse PCR is unsatisfactory because a smaller fragment is always preferentially ligated and amplified [15].

Digestion-ligation-mediated PCR is compromised in specificity and efficiency, attributed mainly to the necessity of using a linker/adaptor/cassette primer. Like the inverse

PCR, digestion-ligation-mediated PCR requires the pretreatment of genomic DNA prior to PCR. Genomic DNA is digested with an endonuclease, followed by ligation with a linker/adaptor/cassette DNA. The DNA of interest is enriched by 2–3 rounds of nested PCRs sequentially driven by the linker/adaptor/cassette primer pairing with nested site-specific primers [10,16]. Apparently, a non-negligible background will inevitably arise from the amplification of irrelevant DNAs by the linker/adaptor/cassette primer [13,17].

Randomly primed PCR is a completely PCR-based method [13,18]. In this PCR, a walking primer partially anneals to some place(s) on the unknown flank in the single non-stringency cycle and afterwards synthesizes a target single-stranded DNA (ssDNA). This target DNA is then amplified by 2–3 successive rounds of PCRs performed by the walking primer pairing with nested site-specific primers [13,18]. Thermal asymmetric interlaced PCR [12], partially overlapping primer-based PCR [7], and wristwatch PCR [13] are types of randomly primed PCR. Nevertheless, undesired DNAs resulting from the walking primer are a big issue. Moreover, these PCRs involve complicated experimental processes or need multiple walking primers [7,12,13].

In this work, a new genome-walking approach, termed semi-site-specific primer PCR (3SP-PCR), is proposed for the rapid amplification of unknown DNA sequences adjacent to a known region. The 3SP-PCR efficiently enriches target DNA while it suppresses non-target ones through the partial overlap between the semi-site-specific primer used in secondary PCR and the corresponding site-specific primer in primary PCR. We confirmed the feasibility of this method by capturing the flanks of the rice hygromycin (*hyg*) gene [7] and *Levilactobacillus brevis* CD0817 glutamic acid decarboxylase genes (*gadA/R*) [19].

## 2. Materials and Methods

### 2.1. Genomic DNA

*L. brevis* CD0817 was cultivated as previously described [20,21]. The genomic DNA of *L. brevis* CD0817 was extracted using the TIANamp Bacteria DNA Kit (Tiangen Biotech Co., Ltd., Beijing, China) according to the manufacturer's guidance. Rice genomic DNA was kindly provided by the lab of Prof. Xiaojue Peng at Nanchang University (Nanchang, China).

### 2.2. Primer

Three outer site-specific primers (I, II, and III) and one inner site-specific primer were selected from each gene. A semi-site-specific primer, with a 3′-overlap (10 nt, Tm value 20–40 °C) and heterologous 5′-part, was designed for each outer primer. The heterologous 5′-part meets the following criteria: (i) the four bases are evenly distributed; (ii) it cannot contribute to forming a primer dimer or hairpin. Any primer has a length of 20–30 nt with a Tm of 60–65 °C. Each primer or primer pair from severe hairpins or dimers is avoided (Table 1).

**Table 1.** Primers used in this study.

| Gene | Primary PCR oSSP | Secondary PCR semi-oSSP | iSSP |
|------|------------------|--------------------------|------|
| *gadA* | I: GGATGCTGCCTTCGGTGGGTTATTT<br>II: GGTTTAGGGTGGATCGTATGGCGT<br>III: ACAACAATGCTGATACGCTGCCAGA | I: ACTCCAACGGCATCCTGGGTTATTT<br>II: TAAGGTCTTCACTGCGTATGGCGT<br>III: TACTTTCCATAACACCGCTGCCAGA | ACGGTTGACTCCATTGCCATTAACT |
| *gadR* | I: TAGCCAACCGTAAACCTGCGTAAAA<br>II: AACTATCACCCCACAACGTCATCTC<br>III: GGATACTGGCTAAAATGAATTAACTCGGAT | I: AGCTGCGATACTCCACTGCGTAAAA<br>II: GGTCCAGCATAGGGTACGTCATCTC<br>III: CATGTAATAACCTCGCTTCATAACTCGGAT | ACCGTTCATAGGCGAAATTGTTTGT |
| *hyg* | I: AAGACCTGCCTGAAACCGAACTGC<br>II: AGTTTAGCGAGAGCCTGACCTATTG<br>III: GGAAGTGCTTGACATTGGGGAGT | I: TTAGAACTGACCCGACCGAACTGC<br>II: CCGCCTAGCCACTGATGACCTATTG<br>III: CTTCTCAGCCTGGATTGGGGAGT | CAAGGAATCGGTCAATACATACATGGC |

Note: The primary PCRs were individually driven by single oSSPs and the secondary PCRs were conducted by the semi-oSSPs individually pairing with the iSSP in the same row. The identical 3′-parts between a semi-oSSP and its corresponding oSSP in the same row are underlined. oSSP: outer site-specific primer; iSSP: inner site-specific primer; *gad*: glutamic acid decarboxylase gene; *hyg*: hygromycin gene.

*2.3. PCR Procedure*

The primary PCR was performed by a single outer site-specific primer using genomic DNA as template. The secondary PCR was performed by the inner site-specific primer and a primer semi-specific to the outer primer using the primary PCR product as a template. In this work, three outer site-specific primers were selected for each gene to perform three parallel primary 3SP-PCRs; thereafter, the three corresponding semi-site-specific primers were individually paired with the inner primer to perform three secondary 3SP-PCRs (Table 1).

The primary PCR mixture (50 μL) contained 1 μL genomic DNA (microbe, 10–100 ng; or rice, 100–1000 ng), 1 × LA PCR Buffer II (Mg$^{2+}$ plus), 0.4 mM each dNTP, 0.2 μM outer site-specific primer, and 2.5 U TaKaRa LA Taq polymerase (TaKaRa, Dalian, China). The secondary PCR mixture (50 μL) comprised 1 μL primary PCR product, 1 × LA PCR Buffer II (Mg$^{2+}$ plus); 0.4 mM each dNTP, 0.2 μM inner site-specific primer, 0.2 μM semi-site-specific primer, and 2.5 U TaKaRa LA Taq polymerase.

The primary PCR consisted of three stages: (i) five moderate high-stringency (60 °C) cycles; (ii) one low-stringency (25 °C) cycle; and (iii) twenty-five moderate high-stringency (60 °C) cycles. The secondary PCR also consisted of three stages: (i) five high-stringency (65 °C) cycles; (ii) one reduced-stringency (40 °C) cycle; and (iii) twenty-five high-stringency (65 °C) cycles. The detailed thermal parameters for 3SP-PCR are shown in Table 2.

**Table 2.** Thermal cycling parameters of semi-site-specific primer PCR.

| Round of PCR | Thermal Parameters | Cycle Number |
|---|---|---|
| Primary | 94 °C, 2 min | |
| | 94 °C, 30 s; 60 °C, 30 s; 72 °C, 3 min | 5 |
| | 94 °C, 30 s; 25 °C, 30 s; 72 °C, 3 min | 1 |
| | 94 °C, 30 s; 60 °C, 30 s; 72 °C, 3 min | 25 |
| | 72 °C, 10 min | |
| Secondary | 94 °C, 2 min | |
| | 94 °C, 30 s; 65 °C, 30 s; 72 °C, 3 min | 5 |
| | 94 °C, 30 s; 40 °C, 30 s; 72 °C, 3 min | 1 |
| | 94 °C, 30 s; 65 °C, 30 s; 72 °C, 3 min | 25 |
| | 72 °C, 10 min | |

*2.4. Agarose Electrophoresis and DNA Sequencing*

The PCR products were separated by 1.5% agarose gel electrophoresis. The dominant band(s) of secondary PCR was recovered using the DiaSpin DNA Gel Extraction Kit (Sangon Biotech Co., Ltd., Shanghai, China) following the operation instruction. The recovered PCR products were directly sequenced by Sangon Biotech Co., Ltd. (Shanghai, China). DNA sequencing was completed by using the ABI sequencer (3730xl DNA Analyzer). The effective sequencing length per reaction is not less than 700 nt, generally up to 850 nt, with an accuracy of 99% or above. The obtained sequences were aligned using the MegAlign tool in the Lasergene package (DNASTAR, Inc., Madison, WI, USA).

## 3. Results

*3.1. Principle of 3SP-PCR*

The principle of 3SP-PCR is indicated in Figure 1. A 3SP-PCR comprises primary and secondary amplification reactions. The key to 3SP-PCR is the use of a semi-site-specific primer in the secondary reaction. This semi-site-specific primer 3′-part overlaps the primer 3′-part used in the primary PCR, while their 5′-parts are completely heterologous to each other (Table 1).

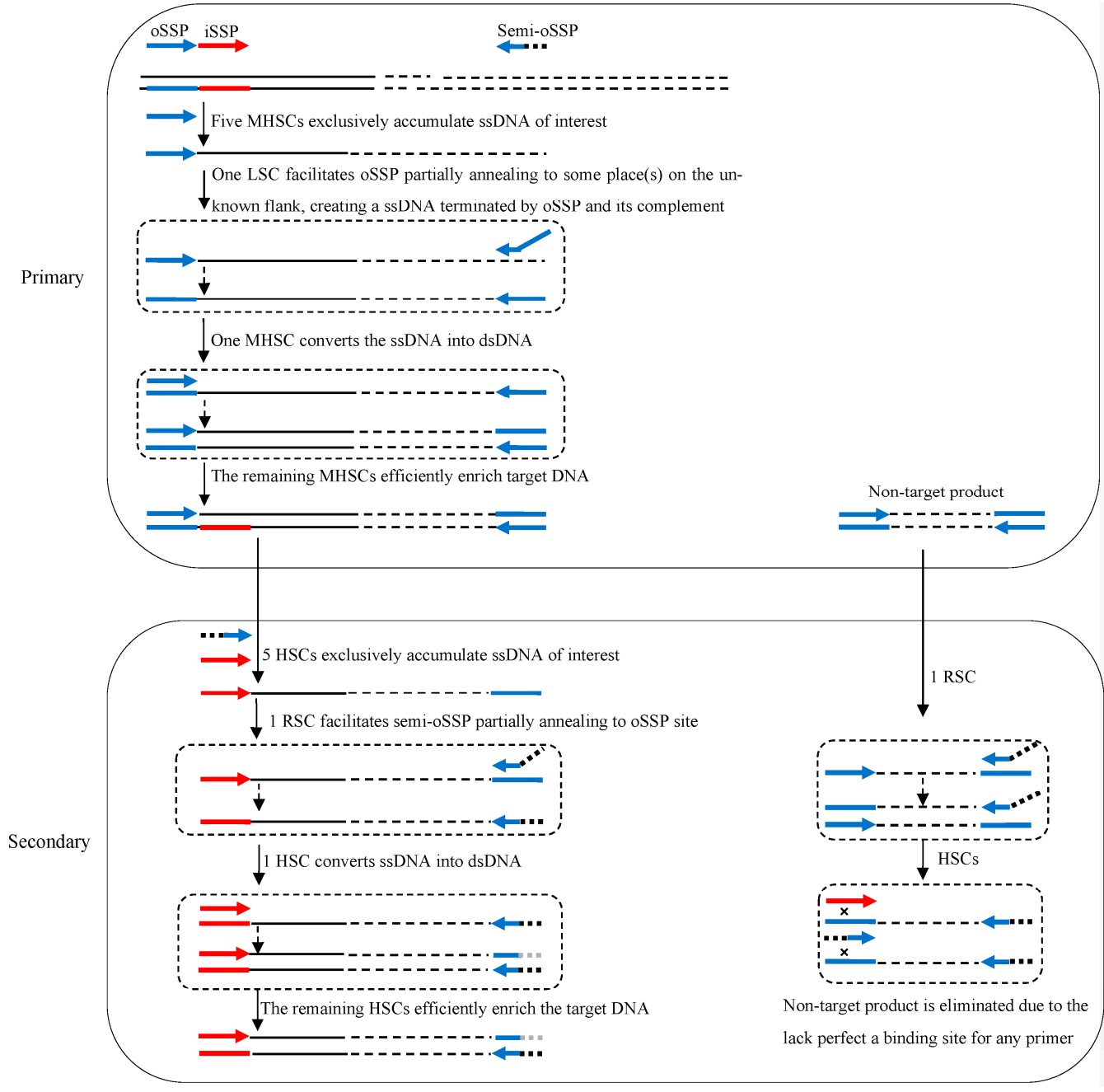

**Figure 1.** Schematic diagram of semi-site-specific primer PCR. oSSP: outer site-specific primer; semi-oSSP: semi-outer-site-specific primer; iSSP: inner site-specific primer. Thin solid line: known sequence; thin dotted line: unknown sequence; colored thick arrows: primers; colored thick lines: primers' complements; ssDNA: single-stranded DNA; dsDNA: double-stranded DNA; MHSC: moderate high-stringency cycle; HSC: high-stringency cycle; LSC: low-stringency cycle; RSC: reduced-stringency cycle.

The primary 3SP-PCR is performed by a single outer site-specific primer. The first five moderate high-stringency (60 °C) cycles exclusively enrich the ssDNA of interest, because the primer can only bind with its complement on the known region. The subsequent one low-stringency (25 °C) cycle allows the primer to partially anneal to numerous sites on genomic DNA or some site(s) on this target ssDNA. A pool of ssDNAs, consisting of target and non-target DNAs, is thereafter newly generated. In the following one moderate high-stringency cycle, the nascent target ssDNA can be converted into double-stranded DNA (dsDNA), because its 3'-end is complementary to the primer. Clearly, this dsDNA

of interest can be efficiently amplified in the remaining moderate high-stringency cycles. Any non-target ssDNA formed in the low-stringency cycle, however, cannot be converted into dsDNA in the following moderate high-stringency cycles because it lacks an authentic binding site of the primer. Hence, the non-target ssDNA cannot be further amplified (Figure 1).

The secondary 3SP-PCR is driven by an inner site-specific primer and a primer semi-specific to the outer primer. The initial five high-stringency cycles only allow the inner primer to bind with its complementary locus on the known region, thus increasing copies of the target ssDNA. The subsequent one reduced-stringency cycle permits the semi-specific primer to partly hybridize to the primary primer locus on target ssDNA, and primes DNA polymerization elongation towards the inner primer site. The complement of the inner primer is thus incorporated into the 3′-end of the target ssDNA. In the following one high-stringency cycle, this ssDNA is converted into dsDNA encompassed by the inner primer and semi-specific primer, which can be exponentially amplified in the remaining high-stringency cycles. The non-target ssDNAs generated in the reduced-stringency cycle cannot be converted into dsDNA due to the lack of a perfect annealing site for any primer. The non-target DNA is therefore diluted in the PCR reaction (Figure 1).

### 3.2. Validation of 3SP-PCR

For a proof-of-concept, we adopted 3SP-PCR to walk the unknown sequences bordering L. brevis CD0817 gadA/gadR genes [19] and the rice hyg gene [7]. Three outer site-specific primers and one inner site-specific primer were designed from each gene, and one semi-specific primer was synthesized for each outer primer (Table 1). Three sets of 3SP-PCRs were then conducted for each gene. The three primary PCRs, using genomic DNA as a template, were individually performed by the three single outer site-specific primers. Then, the three secondary PCRs, using the corresponding primary PCR products as templates, were performed by the three semi-specific primers, respectively, pairing the inner primer. The PCR products were electrophoresed on 1.5% agarose gel. As shown in Figure 2, any secondary 3SP-PCR exhibits 1–2 clear major bands while the background is negligible. These bands were recovered and then sequenced. The data demonstrated that all the bands belong to products of interest (Supplementary Figure S1). The longest distance for each walking constituted by three sets of 3SP-PCRs ranged from 1.5–8.0 kb.

### 3.3. Partial Annealing Sites of Walking Primers

The 3SP-PCR relies on the low-stringency cycle conducted in the primary PCR. This cycle allows the outer site-specific primer (also acts as walking primer) to partially anneal to some place(s) on an unknown region, resulting in the walking of the method. The sequence alignments between the outer site-specific primers and their partial binding loci are summarized in Figure 3. The results illustrated that an effective partial annealing contains at least two matched base pairs at the primer 3′ end. Moreover, the total matched base pairs for a primer ranged from 8 to 17.

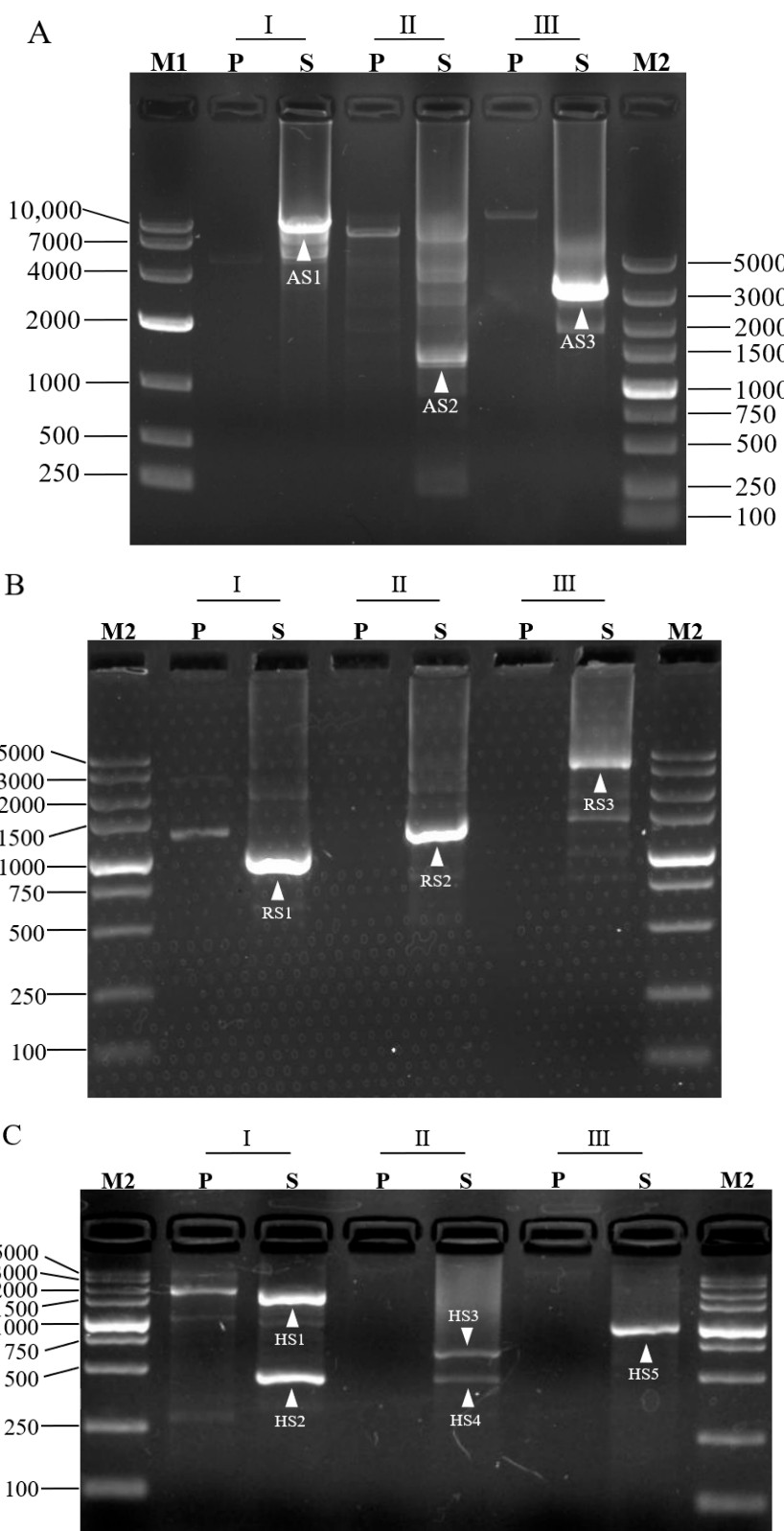

**Figure 2.** Probing upstream of *gadA* gene (**A**), downstream of *gadR* gene (**B**) of *L. brevis* CD0817, and downstream of *hyg* gene (**C**) of rice. Marks I, II, and III denote the three parallel sets of 3SP-PCRs in each walking experiment. P: primary PCR; S: secondary PCR; M1: DNA 10,000 Marker (TaKaRa, Dalian, China); M2: DNA 5, 000 Marker (TaKaRa, Dalian, China). The fragments AS1-AS3, RS1-RS3, and HS1-HS5 are the secondary PCR amplicons of gadA, gadR, and hyg, respectively.

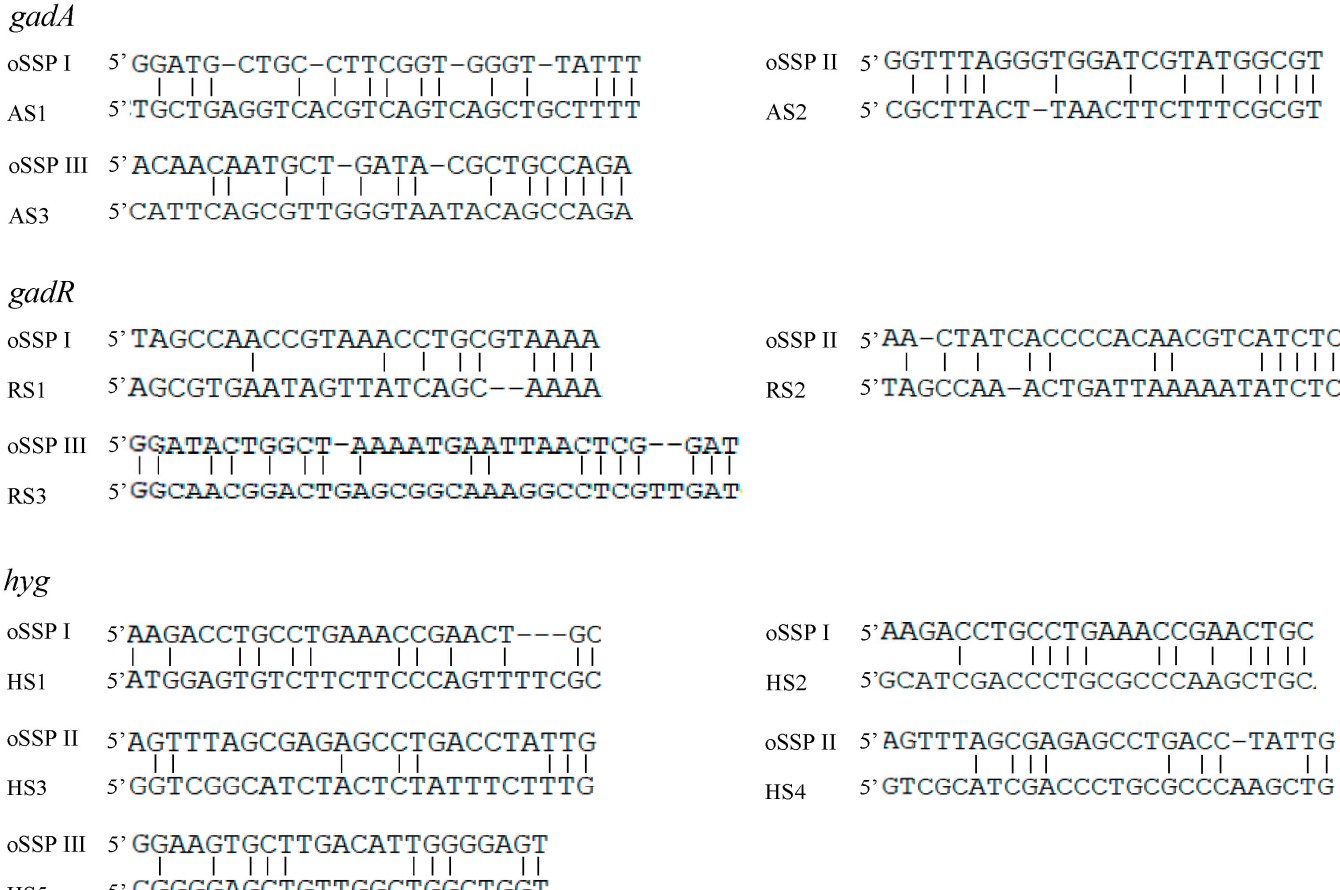

**Figure 3.** Partial annealing locations of the oSSP primers. The DNA bands AS1-AS3, RS1-RS3, and HS1-HS5 correspond to the ones described in Figure 2.

### 3.4. Discussion

To date, many genome-walking methods, related to DNA library screening, inverse PCR, digestion-ligation-mediated PCR, or randomly primed PCR, have been developed [8,11,13,22]. Among which, randomly primed PCR-based genome-walking methods, such as thermal asymmetric interlaced PCR [12], partially overlapping primer-based PCR [7], and stepwise partially overlapping primer-based PCR [15], have features of rapidity and simplicity. However, these methods are unsatisfactory owing to their limited level of efficiency, high background, or complex operations [13]. Therefore, it is attractive to develop a randomly primed PCR-based genome-walking method of real practicability. The proposed 3SP-PCR is such a genome-walking protocol.

The 3SP-PCR is of high specificity because it owns the following two specificity safeguard mechanisms. First, the relatively short (10 nt) 3′ overlap, between the semi-specific primer and its corresponding outer site-specific primer is the key to guarantee the specificity. In each round of 3SP-PCR, the single relaxed-stringency cycle limits the partial annealing of any primer to once only, and produces a pool of ssDNAs. The non-target ssDNA cannot be amplified in the subsequent moderate high-stringency cycles, as it lacks an authentic binding site for any primer. The target one, however, can be amplified in the subsequent moderate high-stringency cycles, as its 3′-end has the binding site for the site-specific primer (Figure 1). In this regard, 3SP-PCR possesses a specificity safeguard mechanism similar to that of stepwise partially overlapping primer-based PCR [7,15] or wristwatch PCR [13]. Second, types of non-target DNAs are reduced in 3SP-PCR. The existing randomly primed PCRs have a common feature; that is, they all have to use a random walking primer. Clearly, three types of non-target products are thus produced in these methods: (i) produced from a random walking primer alone; (ii) produced from

random walking primer and site-specific primer; and (iii) produced from site-specific primer alone [1,12,13]. Our 3SP-PCR does not involve a random walking primer given that the primary PCR is a single-primer PCR, in which the outer site-specific primer simultaneously acts as a walking primer. Hence, types i and ii non-target molecules are absent from the amplified product. This is an extra mechanism exclusive to 3SP-PCR for removing non-target background.

The 3SP-PCR is simpler than the available randomly primed PCRs. As discussed above, three types of non-target products are formed in routine randomly primed PCRs primary amplification [1,12,13]. Therefore, three rounds of amplification reactions are generally required in these methods, so as to dilute non-target background to an acceptable level [1,12]. For the proposed 3SP-PCR, two rounds of amplification reactions suffice to give a positive outcome (Figure 2), because its primary amplification generates only one type of non-target DNA (Figure 1).

The multi-band phenomenon predominated by short amplicons, common in routine randomly primed PCRs [1,13,15,23], is significantly ameliorated in 3SP-PCR. In general, a walking primer should identify numerous sites on the unknown flanking DNA under the low temperature (25 °C). As a result, the multi-band phenomenon occurs [7,15]. In addition, a smaller fragment is always preferentially amplified because it has a higher amplification efficiency than that of a longer one. Therefore, small amplicons seem to be predominant in traditional randomly primed PCRs [12,13]. In the primary 3SP-PCR, amplicons are encompassed by the site-specific primer and its inverted repeat. The amplification of a shorter fragment is inhibited because it is a preferentially formed hairpin via intra-strand annealing at the both termini. However, the amplification of a longer DNA is much less inhibited [24]. As shown in Figure 2, each secondary 3SP-PCRs exhibits only 1–2 distinct bands, demonstrating the multi-band phenomenon is improved in this method. Moreover, the largest fragment achieved in each walking ranged from nearly 1.5 to 8.0 kb, indicating a satisfactory walking efficiency of 3SP-PCR.

The success rate of 3SP-PCR is also guaranteed. The walking should succeed as long as the primer partially anneals to the unknown flank in the single low-stringency (25 °C) cycle of primary PCR [7,12,13,15]. We suspect that, at such a low temperature, the primer can find at least one site on the flank suitable for partial annealing, not to mention multiple primers can be individually used in parallel. In this work, one walking comprising three 3SP-PCR sets was performed for each of the selected genes. As shown in Figure 2, all 3SP-PCRs generated positive results, indicating a high success rate of the 3SP-PCR. A comparison of 3SP-PCR with the existing randomly primed PCRs is presented in Table 3.

Reportedly, a primer can work even if the base mismatch rate is up to 60% [25]. In this study, the matched bases between the outer site-specific primer and their partial annealing loci were from 8 to 17, with at least two matched bases at the primers 3′ ends (Figure 3), which is accordant with the reported data [24–26]. At least two bases pairing at the 3′ end should be essential for the functional priming of a primer [24–26].

**Table 3.** Comparison of 3SP-PCR with the available PCR-based methods.

| Approach | Rationale and Process | ESGM | Walk Range | Reference |
|---|---|---|---|---|
| Inverse PCR | Genomic DNA is digested with an endonuclease then self-circularized. The resultant DNA undergoes PCR performed by two SSPs with opposite extension orientations. | No | 0.3–1.8 | [1,8] |
| DL-PCR | DNA is digested with an endonuclease then ligated to a random oligo. The resultant DNA undergoes 2–3 consecutive PCRs by nested pairing the oligo primer with SSPs. | No | 0.3–3.0 | [1] |
| TAIL-PCR | A short random oligo is used as a walking primer. One low-stringency cycle has to be performed in every three cycles to aid walking primer annealing. The amplification of a target DNA is over a non-target one due to the differential amplification efficiency. | Yes | 0.3–3.5 | [1,12] |
| POP-PCR | A set of walking primers with 3′-overlaps are individually paired with nested SSPs. A walking primer can partially anneal to a DNA template only in the one relaxed-stringency cycle of each PCR, generating a pool of ssDNAs. A target ssDNA can be converted into dsDNA by the SSP in the next high-stringency cycle, but a non-target one cannot. | Yes | 0.3–3.5 | [7] |
| 3SP-PCR | The rationale and process are shown in the section of "3.1 Principle of 3SP-PCR" of this study. | Yes | 0.4–8.0 | This study |

Note: SSP denotes site-specific primer. DL-PCR, TAIL-PCR, POP-PCR, WW-PCR, and 3SP-PCR represent digestion-ligation-mediated PCR, thermal asymmetric interlaced PCR, partially overlapping primer-based PCR, wristwatch PCR, and semi-specific-primer PCR, respectively. ESGM: extra success-guaranteeing mechanism.

## 4. Conclusions

In this study, a simple, rapid but reliable 3SP-PCR has been devised for genome-walking. The feasibility of the proposed method was validated by retrieving the unknown flanking DNA sequences in rice and a microbe. The 3SP-PCR should be a promising alternative to the available genome-walking approaches.

**Supplementary Materials:** The following supporting information can be downloaded at: https://www.mdpi.com/article/10.3390/cimb45010034/s1, Figure S1: Alignments of the PCR products.

**Author Contributions:** C.W.: investigation, visualization, and writing—original draft preparation; Z.L.: data collection, J.P.: analyzation, H.P.: resource provision; H.L.: conceptualization, funding acquisition, and writing—review and editing. All authors have read and agreed to the published version of the manuscript.

**Funding:** This work was funded by the National Natural Science Foundation of China (grant No 32160014), and the State Key Laboratory of Food Science and Technology of Nanchang University (grant No SKLF-ZZB-202118).

**Institutional Review Board Statement:** Not applicable.

**Informed Consent Statement:** Not applicable.

**Data Availability Statement:** Not applicable.

**Acknowledgments:** The authors gratefully thank Xiaojue Peng of Nanchang University (Nanchang, China) for providing us with the rice genomic DNA.

**Conflicts of Interest:** The authors declare no conflict of interest.

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
