# Peer review of "Semi-Site-Specific Primer PCR: A Simple but Reliable Genome-Walking Tool"

_cimb, doi:10.3390/cimb45010034_

Round 1

Author Response

Reviewer 1

Point 1: What is the amplification efficiency of this method? Is it effective for low-copy genes?

Response: Thank you for this comment. In general, it is unnecessary to determine the amplification efficiency of a genome-walking method. The exact amplification efficiency of 3SP-PCR is unclear. However, its amplification efficiency should be comparable to that of the classical end-to-end PCR, because 3SP-PCR amplifies a target DNA like the classical end-to-end PCR.

The 3SP-PCR should be effective for low-copy genes, because we used the single-copy genes to validate the method.

Point 2: Whether the method can be used in the mRNA to amplify the UTR region of the gene?

Response: Thanks a lot. Currently, the 3SP-PCR is not a good method used in mRNA to amplify the UTR region of a gene. The 3SP-PCR amplifies a DNA by random annealing on the unknown flank. Therefore, the 3SP-PCR may not obtain the complete UTR region of the gene.

Point 3: Primer design of Semi site-specific primer PCR needs to be written more succinctly.

Response: Thank you for this comment. We have accordingly revised this paragraph as follows: Three outer site-specific primers (I, II, and III) and one inner site-specific primer were selected from each gene. A semi site-specific primer, with a 3’-overlap (10 nt, Tm value 20-40 ℃) and heterologous 5’-part, was designed for each outer primer. The heterologous 5’-part meets the following criteria: (i) the four bases are evenly distributed; and (ii) cannot contribute to forming primer dimer or hairpin. Any primer has a length of 20-30 nt with a Tm of 60-65℃. Each primer or primer pair is avoided from severe hairpins or dimers (Table 1).

Point 4: All of the tables in your paper should be changed to the standard three-line

table.

Response: Thank you for this comment. All the tables have been revised accordingly.

Point 5: Please check on your reference format carefully and modify it. Such as reference 7, 11, 13, 18, and so on.

Response: Yes, we have thoroughly checked the references and revised accordingly.

Reviewer 2 Report

In this manuscript, Wei et al. presented a simple, rapid and reliable approach for genome walking, semi site-specific primer PCR (3SP-PCR) which includes two rounds of nested amplification reactionsThe feasibility was also validated by retrieving the unknown flanking DNA sequences in rice and a microbe. The research was designed correctly, the results are interpreted well. The manuscript could be accepted. 

Author Response

Reviewer 2

In this manuscript, Wei et al. presented a simple, rapid and reliable approach for genome walking, semi site-specific primer PCR (3SP-PCR) which includes two rounds of nested amplification reactions. The feasibility was also validated by retrieving the unknown flanking DNA sequences in rice and a microbe. The research was designed correctly, the results are interpreted well. The manuscript could be accepted. 

Response: Thank you efforts on our manuscript.